# Open reproducible scientometric research with Alexandria3k

**Diomidis Spinellis**[ID][1,2]*

**1** Department of Management Science and Technology, Athens University of Economics and Business, Athens, Greece, **2** Department of Software Technology, Delft University of Technology, Delft, The Netherlands

* dds@aueb.gr

**Data Availability Statement:** A versioned release of the source code of Alexandria3k and the proof-of-concept examples presented in this study is available on Zenodo doi: 10.5281/zenodo.7953287, licensed under the GNU General Public License v3.0. A replication package with the results data

## Abstract

Considerable scientific work involves locating, analyzing, systematizing, and synthesizing other publications, often with the help of online scientific publication databases and search engines. However, use of online sources suffers from a lack of repeatability and transparency, as well as from technical restrictions. Alexandria3k is a Python software package and an associated command-line tool that can populate embedded relational databases with slices from the complete set of several open publication metadata sets. These can then be employed for reproducible processing and analysis through versatile and performant queries. We demonstrate the software's utility by visualizing the evolution of publications in diverse scientific fields and relationships among them, by outlining scientometric facts associated with COVID-19 research, and by replicating commonly-used bibliometric measures and findings regarding scientific productivity, impact, and disruption.

## Introduction

Research synthesis is becoming an increasingly important [1] and popular scientific method. A major data source for research synthesis studies are online specialized and general purpose bibliographic and article databases [2], such as Dimensions [3], ERIC [4], Google Scholar [5], Inspec, OpenAlex [6], Scopus [7], and Web of Science [8]. These databases can be used through their online interfaces [2] or via diverse (historic or actively maintained) visualization and analysis software packages and tools that can import their data [9, 10], such as bibliometrix [11], CitNetExplorer [12], CorTexT Manager [13], HistCite [14], pySciSci [15], Sci² [16], and VOSviewer [17].

Performing systematic studies on published literature through the available online systems or proprietary datasets can be problematic [18]. First, their constantly updated contents, bubble effects [19], location– and license-dependent results [20], and periodic changes to their internal workings compromise reproducibility [21, 22]. Even when the search strategy is well-documented to aid reproducibility by following recommended reporting guidelines such as PRISMA [23], which is often not the case [24, 25], it is difficult to repeat a query to an online service, and obtain the same results as those that have been published [18]. In addition, the reproducibility of such studies is hampered in the short term by the fees required for accessing

and Python scripts used for the statistical analysis and charting of the reported example studies is also available on Zenodo doi: 10.5281/zenodo. 8273279. Current versions of Alexandria3k are made available for installation through PyPI https:// pypi.org/project/alexandria3k/ and for contributions, feature requests, and issue reporting on GitHub https://github.com/dspinellis/ alexandria3k. The data used in the example studies are available as follows. Crossref Apr. 2022 Public Data File: doi:10.13003/83b2gq ORCID Public Data File 2022 v. 4: doi:10.23640/07243.21220892.v4 ROR Data v1.17.1: doi:10.5281/zenodo.7448410 Open access journals: https://doaj.org/csv doi:10. 5281/zenodo.8265889 Funders: https://doi. crossref.org/funderNames?mode=list doi:10.5281/ zenodo.8265889 Journals: http://ftp.crossref.org/ titlelist/titleFile.csv doi:10.5281/zenodo.8265889 The calculated $CD_5$ index of works published in the period 1945-2016 dataset is available on Zenodo doi: 10.5281/zenodo.7679112. The 2021 Journal Impact Factor data used for assessing the numbers obtained by Alexandria3k are available from Clarivate through the InCites Journal Citation Reports service https://jcr.clarivate.com/.

**Funding:** The Alexandria3k has been supported by the Google Summer of Code 2023 program via the Open Technologies Alliance. The funders had no role in study design, data collection and analysis, decision to publish, or preparation of the manuscript.

**Competing interests:** Google Summer of Code through the Open Technologies Alliance supported the work of a student developer on Alexandria3k. The author has declared that no competing interests exist. This does not alter our adherence to PLOS ONE policies on sharing data and materials.

many online services, and in the long term by the commercial survival of the corresponding companies [26]. Service access costs on their own can restrict institutions with limited funding from conducting systematic literature studies. Another associated problem is the lack of transparency [27]. Most online services work with proprietary data collections and algorithms, making it difficult to understand and explain the obtained results. As an example, Clarivate's journal impact factor calculation depends on an opaque collection of journals [28] and list of "citable items" [29] tagged so by its vendor. Finally, there are technical limitations. Some services lack a way to access them programmatically (an application programming interface— API) [18], forcing researchers to resort to tricky and unreliable contortions, such as screen scrapping. Both APIs and offered query languages are not standardized [30], and often restrict the allowed operations [18]. For instance, the network-based APIs suffer from corresponding latency [31], and often from rate and ceiling limits to the number of allowed invocations [32, 33]. These restrictions hinder studies requiring a large number or complex queries.

The outlined problems can be addressed thanks to sustained exponential advances in computing power [34], drops in associated costs, and Open Science initiatives [35]. The Alexandria3k system presented here is an open-source software library and command-line tool that builds on these advances to allow the conduct of sophisticated systematic research of published literature, (e.g. literature reviews, meta-analyses, bibliometric and scientometric studies) in a transparent, repeatable, reproducible [36], and efficient manner. Alexandria3k allows researchers to process on a personal computer publications' metadata (including citations) from most major international academic publishers as well as corresponding author, funder, organization, and journal details. Specifically, Alexandria3k works on data snapshots offered periodically by initiatives, such as Crossref (publication metadata, journal names, funder names) [37], ORCID (author details) [38], ROR (research organization registry) [39], and others. Using Alexandria3k researchers can query and process these data through SQL queries launched by means of command-line tool invocations or Python scripts. Researchers can ensure the transparency, reproducibility, and exact repeatability of their methods by documenting or publishing (when permitted) the version of the data used and the employed commands [35]. All examples provided in this paper are made available online in this manner.

The architecture and features of Alexandria3k allow most tasks to be executed on a modern personal computer. Specifically, Alexandria3k offers facilities for running relational database queries on compressed JSON data partitions, for sampling records on-the-fly, and for populating a relational database with a subset of a large dataset's records or fields. At no point are the voluminous primary data decompressed or loaded into main memory in their entirety.

Consequently, in contrast to most other related software systems, Alexandria3k requires only modest computing resources. Regarding secondary (disk) storage, the primary data sets can be stored and processed locally, because they amount to a few hundred gigabytes in their compressed format (157 GB for Crossref, 25 GB for ORCID data; the downloading of the Crossref data is facilitated by its availability through the BitTorrent protocol [40]). The data are decompressed in small chunks thus curbing both main and secondary memory requirements. (Keeping the data decompressed or populating a relational or graph database with all of it would require more than 1.5 TB of storage space.) From an execution time perspective, on a populated and suitably indexed database with millions of records, many queries finish in minutes. Queries or database population tasks involving a full scan of the entire Crossref publication data set complete in about five hours. From a main memory (RAM) perspective, the database population and query tasks require less than a GB of RAM. (Full performance details are provided in the Methods section.) In contrast, pySciSci [15], which shares goals, data, and features with Alexandria3k, but is designed around Python data frames, requires amounts of main memory (RAM) that are not typically found on personal devices. As an example, we

calculated that preprocessing with pySciSci the publications and references of the OpenAlex [6] database (equivalent to Alexandria3k's use of the Crossref data set) would require more than 81 GB of RAM only for the works.

## Contents, structure, and use

In total, Alexandria3k offers relational query access to 2.6 billion records. These are organized in a relational schema illustrated in S1–S4 Figs.

Most records are publication metadata obtained from the Crossref Public Data File [37]. These contain publication details (DOI, title, abstract, date, venue, type, pages, volume, . . .), a publication's references to other publications (DOI, title, author, page, ISSN, ISBN, year, . . .) [41], and other data associated with each publication's authors and their affiliations, funders and funder awards, updates (e.g. retractions), subjects, licenses, and hyperlinks for text mining of the publication's full text [37]. Details about the data available through Crossref are listed in Table 1. Note that coverage is incomplete; for example, 39% of the publications have a reference list associated with them, 70% of funders are uniquely identified with a DOI, while only 11% of the publications have an abstract. For most types of records coverage is generally increasing over time (see Fig 1).

Alexandria3k can unambiguously link Crossref records to imported author metadata through ORCID (Open Researcher and Contributor ID): a non-proprietary system developed to identify authors in scholarly communication [38]. ORCID tables that Alexandria3k supports include those detailing an author's URLs, countries, keywords, external identifiers, distinctions, education, employment, invited positions, memberships, qualifications, services, fundings, peer reviews, used research resources, and published works. Most of these tables

**Table 1. Number and properties of Crossref records.**

| Entity | Records |
|---|---:|
| Total records | 2 531 227 295 |
| Works (publications) | 134 048 223 |
| Works with a text mining link | 96 294 821 |
| Works with subject | 81 210 089 |
| Works with references | 52 907 361 |
| Works with affiliation | 16 833 863 |
| Works with an abstract | 15 367 820 |
| Works with funders | 7 519 462 |
| Author records (linked to works) | 359 556 891 |
| Author records with ORCID | 16 745 506 |
| Distinct authors with ORCID | 4 525 906 |
| Author affiliation records | 76 768 648 |
| Distinct affiliation names | 19 453 360 |
| Work subject records | 182 858 177 |
| Distinct subject names | 340 |
| Work funders | 15 491 915 |
| Funder records with DOI | 10 811 496 |
| Distinct funder DOIs | 29 610 |
| Funder awards | 14 090 597 |
| References | 1 748 421 617 |
| References with DOI | 1 255 033 889 |
| Distinct reference DOIs | 59 127 679 |

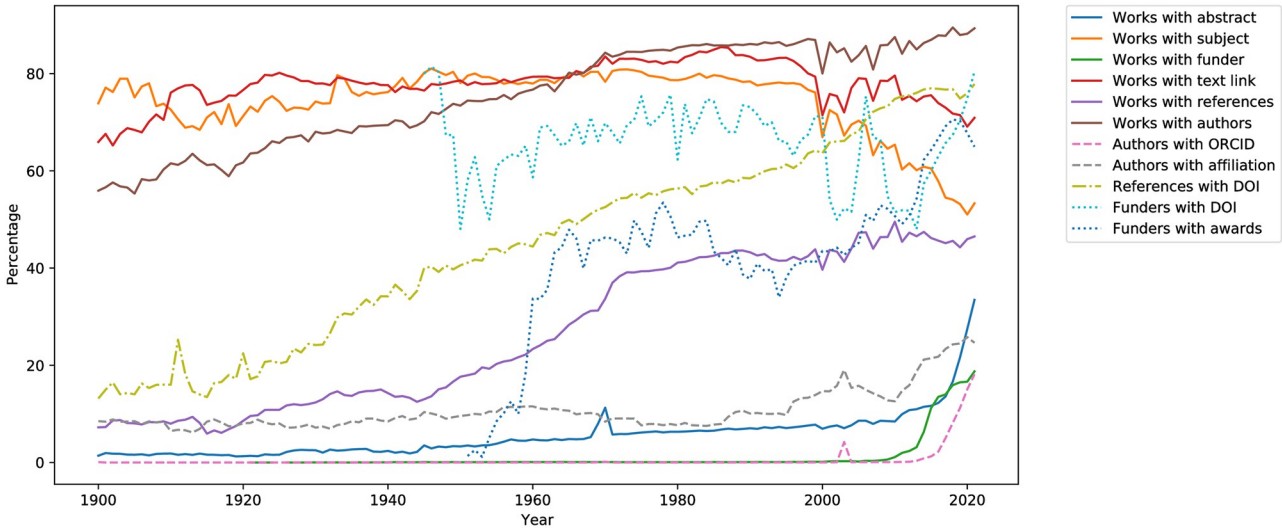

**Fig 1. Availability of Crossref data elements over the period 1900–2021.** Text link availability refers to full-text mining links. Availability of author ORCID and affiliations is evaluated over all of each year's individual author records appearing alongside publications. Similarly, availability of funder DOI and award data is evaluated over all of each year's individual funder records appearing alongside publications. This means e.g. that if an author appears in one publication with an ORCID and in another without, the reported ORCID availability for the corresponding year will be 50%. Availability of funder, text link, references, authors, affiliation, and awards refers to the existence of at least one such record associated with the parent entity. DOI availability in references and funders is evaluated for each individual record of the many associated with one publication.

contain details of the associated organization (name, department, city, region, country), the role title, and the starting and end date. The currently available ORCID data set contains about 78 million records associated with 14 million authors. The completeness of the ORCID records is low and uneven (see Table 2), which means that research based on it must be carefully designed.

**Table 2. Number of ORCID records.**

| Table | Records | Persons with Such Records |
|---|---|---|
| Personal data | 14 811 567 | 14 811 567 |
| URLs | 1 325 399 | 892 528 |
| Countries | 2 141 021 | 2 021 218 |
| Keywords | 2 230 569 | 838 796 |
| External identifiers | 2 287 063 | 1 653 995 |
| Distinctions | 345 967 | 159 113 |
| Educations | 5 610 051 | 3 087 381 |
| Employments | 5 932 529 | 3 472 114 |
| Invited positions | 263 315 | 147 497 |
| Memberships | 660 304 | 357 799 |
| Qualifications | 940 400 | 585 298 |
| Services | 221 860 | 108 288 |
| Fundings | 1 170 432 | 328 958 |
| Peer reviews | 6 270 131 | 751 404 |
| Research resources | 3 012 | 1 779 |
| Person's works | 34 237 877 | 3 224 068 |
| Total | 78 451 497 | |

Alexandria3k can also import the Research Organization Registry data [39] containing details of 104 402 organizations, as well as related acronyms (43 862 records) and aliases (25 119 records). Through the provided ROR identifier it can link unambiguously elements from a person's employment and education ORCID records to the corresponding organization. Currently ORCID contains such identifiers for 130 033 employment records and 133 066 education records. Because only 4.6% of work author records have an ORCID and only 23% of ORCID records contain employment information, Alexandria3k also provides a performant facility to match the textual affiliation information listed in works and link it to ROR identifiers.

Finally, Alexandria3k can import and link three reference tables: the names of journals associated with ISSNs (currently 109 971 records), the Research Organization Registry (ROR) containing funder names associated with funder DOIs (32 198 records), and the Directory of Open Access Journals (DOAJ; 18 717 records) [42]. Alexandria3k further disaggregates journal ISSNs according to their type (electronic, print, or alternative—158 580 records).

The data used by Alexandria3k is openly distributed [43–46] by diverse third parties in textual tree or flat format files: JSON for Crossref and ROR, XML for ORCID, and CSV for the rest. Alexandria3k structures the data it offers in a relational schema of 45 tables linked through 47 relations. Stored in a relational database and combined with suitable indexes, this allows performing sophisticated analyses via SQL (structured query language) queries in an efficient manner. Records between diverse data sets are linked through standardized globally unique identifiers: DOIs for published works and funders, ISSNs for journals, ORCIDs for authors, and RORs for research organizations.

Alexandria3k is distributed as open source software in the form of a Python package, which can be easily installed through the PyPI repository. It can be used either as a command-line tool, where its operation (e.g. query to run) is specified through command-line arguments, or as a Python library, which can be used interactively (e.g. by developing a Jupyter Notebook [47]) or through scripts. The performance of both interfaces is similar, because in both cases the heavy lifting is done by the same Python library code.

The 2022 version of the Crossref data reported in this study is distributed as 26 thousand compressed container files, each containing details about 5 000 works. A complete import of the Crossref data would amount to a 520 GB database. Given the large amount of Crossref data, the population of a database with it can be controlled in three ways. First, only a horizontal subset of records can be imported, by specifying an SQL expression that will select a publication record only when it evaluates to `TRUE` (e.g. `published_year BETWEEN 2017 AND 2021`). To facilitate the selection of records selected through other means, the expression can also refer to tables of external databases. Second, only a subset of the Crossref 26 810 data containers can be processed for import, by specifying a Python function that will import a container when it evaluates to `True`. This is mostly useful for random sampling, e.g. using `random.random() < 0.01` to sample approximately 1% of the containers. (A fixed seed value is used internally for initializing the pseudo-random number generator to allow deterministic and therefore repeatable sampling.) Third, the populated tables or columns of the Crossref data set can be vertically restricted by using the `table-name.column-name` or `table-name.*` SQL notation. The population of a database with ORCID data can be also horizontally restricted to records associated with existing Crossref authors or published works (probably selected in a previous population step) and vertically restricted to include only specific tables or columns, as in the case of Crossref. Given their small volume, no population controls are supported for the other data sets.

The analysis of the extracted data can be performed either directly with SQL queries and *ad hoc* code, as shown in the next section, or by utilizing specialized software. For example, once

Alexandria3k has populated a database, its data can processed in R [48] using the *bibliometrix* tool for science mapping analysis [11]. This can be done by creating R data frames or tibbles (enhanced data frames provided by the *tidyverse* package) from data brought into R through its *DBI* and *RSQLite* packages.

## Application examples

The following paragraphs outline some simple proof-of-concept applications of Alexandria3k, which demonstrate its use and motivate its adoption. All are exactly replicable through SQL queries and relational online analytical processing (ROLAP) Makefiles [49–51] provided in the accompanying materials.

### Evolution of scientific publishing

Bibliometric indicators are often used to measure scientific progress, evolution, and research [52, 53]. Fig 2 showcases the use of Alexandria3k to chart a view of scientific publishing evolution in the post-WW2 period. It shows an exponential rise in the number of published works (CAGR—compound annual growth rate = 5.4%; $R^2$ = 0.99) and in the number of journals (CAGR = 5.1%; $R^2$ = 0.99), in line with what is reported by the trade association study [54, p. 27]. Despite this rise, publications are becoming ever more connected by citing each other. This can be seen in the rises of the references each work contains, the citations works receive, the phenomenal proportion of all works ever published that are cited at least once every year (20%), and corresponding rises to the 2-year and even 20-year global impact factor. Authors appear to be collaborating more and on longer papers with only a slight decrease in the mean number that they publish each year. Papers published in journals that are open access at publication time have risen exponentially (CAGR from 1990 = 0.23%; $R^2$ = 0.99) to reach 22% of all journal papers, a ratio that is also mirrored by a published figure [54, p. 8]. The fall in the consolidation/destabilization (CD) index is in line with recently published research reporting that

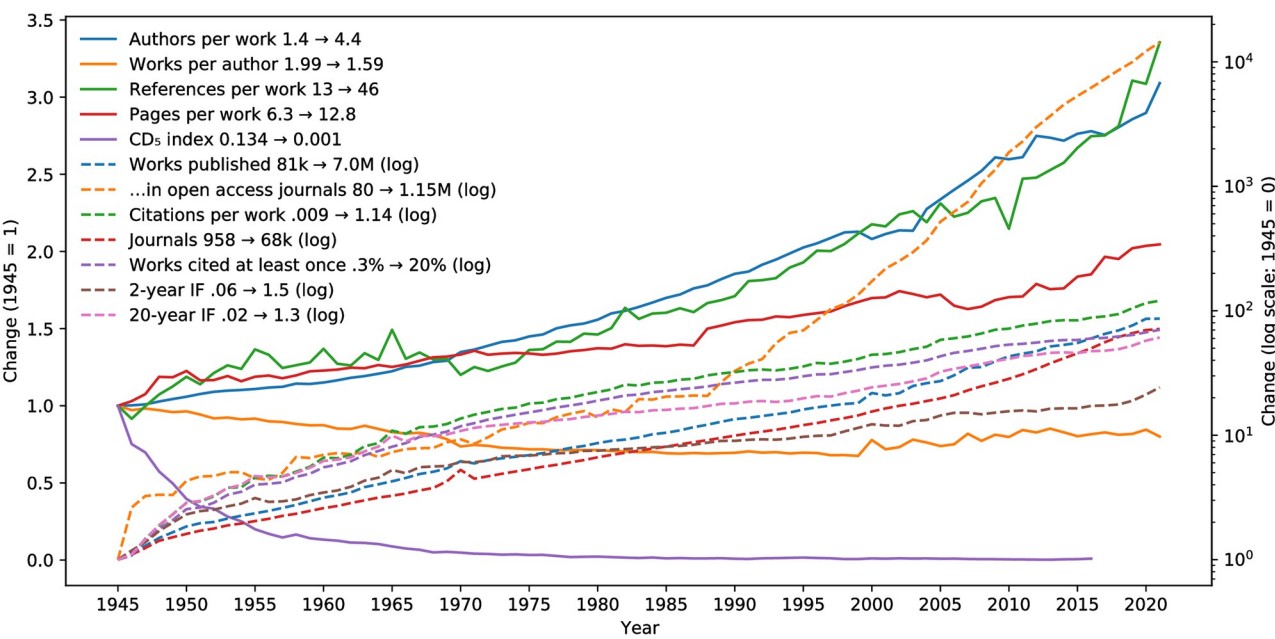

**Fig 2. Evolution of scientific publishing metrics in the post-WW2 period.**

papers are becoming less disruptive over time [55]. There is significant correlation (Spearman rank-order correlation coefficient 0.95; $P < 0.001$) between the $CD_5$ index yearly averages obtained using Alexandria3k with those available in the previously published dataset, which was obtained from data that are not openly available. The sharp inflections in the Figure probably stem from artifacts of the underlying data set, and indicate that in some cases obtaining scientifically robust results might require deeper analysis of the data.

A trend related to this work is the rising number of research synthesis studies. We found about 437 thousand scientific studies published from 1846—the year of the first one we found [56]—onward based on the analysis of previously published primary studies. The number of synthesis studies published each year has risen considerably over the past two decades, particularly for systematic literature reviews (see Fig 3). These were typically identified by their titles through terms such as: "systematic review", "systematic literature review", or "systematic mapping study" (secondary studies using methods that help make their findings unbiased and repeatable—251 850 titles); "secondary study", "literature survey", or "literature review" (a not necessarily systematic study reviewing primary studies—77 037 titles); "tertiary study" or "umbrella review" [57] (a study reviewing secondary studies—4 039 titles); "meta-analysis" [58] (a systematic secondary study employing statistical methods—92 363 titles); as well as (systematic by definition) "scientometric" (employing quantitative methods to study scientific research—2 769 titles), "bibliometric" (using statistical methods to study written communications—12 361 titles), and broader Science of Science (SciSci—studies of scientific processes based on large data sets) [59] studies [58].

The size of scientific fields can affect canonical progress [60] while the corresponding size dynamics can be associated with a field's evolutionary stages [61]. Associating publications with the general scientific field of the journal they were published (according to the Scopus All Science Journal Classification Codes—ASJCs) allows Alexandria3k to generate a view of the

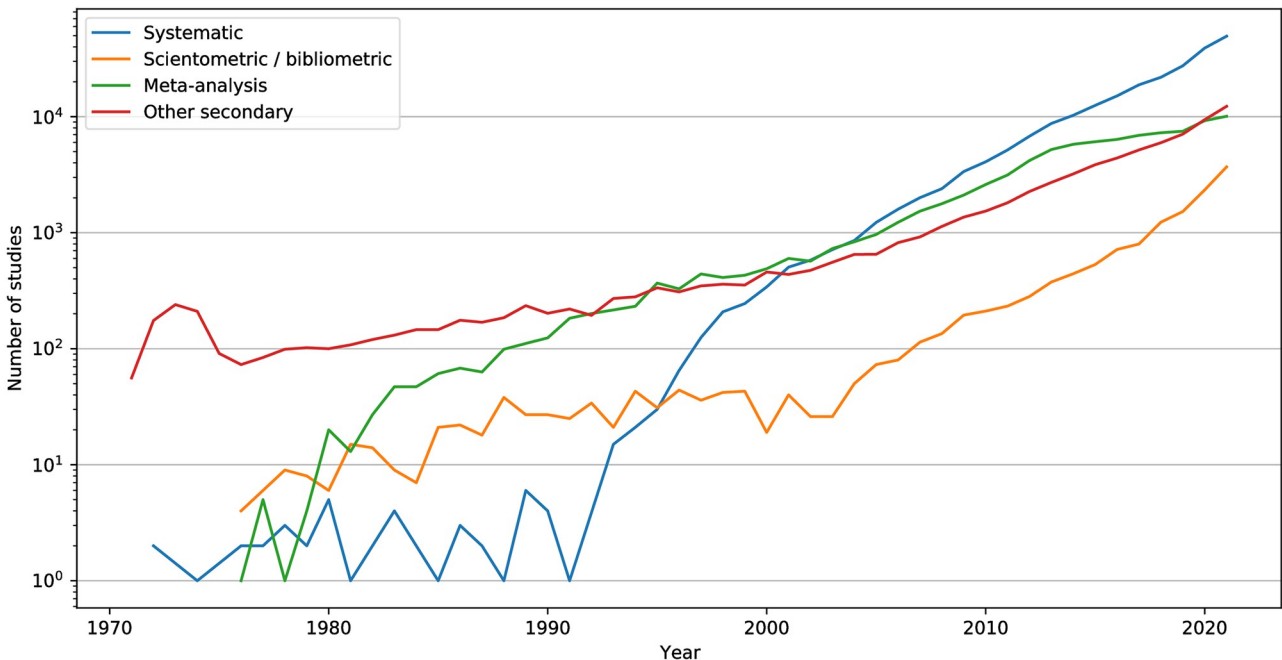

**Fig 3. Number of research synthesis studies published each year in the period 1971–2021.**

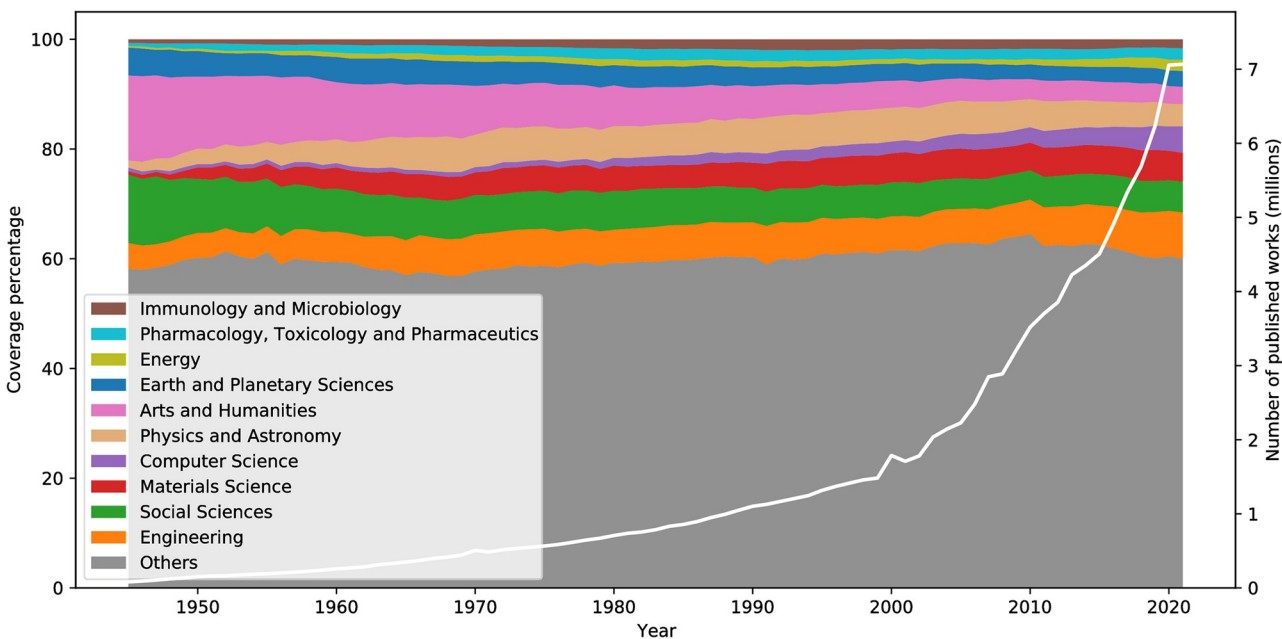

**Fig 4. Evolution of subject coverage and publications 1945–2021.**

evolution of the ASJC 27 fields over the years. Fig 4 shows the ten fields amounting to more than 2% of publications in 2021 that had the largest change in their publication number in the period 1945–2021. Clearly visible is the expected rise in Computer Science, Materials Science, and Physics and Astronomy, as well as the fall of Arts and Humanities and Social Sciences publications. Owing to the exponential rise of published works (tallied on the Figure's right-hand axis) the falls are only in relative terms: 2021 saw the publication of 291 366 Arts and Humanities articles against 19 873 in 1945.

## COVID-19 research

The bibliometric study of a specific scientific topic, such as COVID-19 [62], can aid progress by providing details and insights on the state of the art and the community working on it [63, 64]. To showcase how Alexandria3k could help study research related to COVID-19 [65] we used Alexandria3k to create and study a data set of publications containing COVID in their title or abstract. Our findings are in broad agreement with studies using different research methods [66, 67].

We counted 491 945 publications from about 1.5 million authors. These covered 331 different topics demonstrating the many disciplines associated with the research. Some noteworthy topics and work numbers among those with more than one thousand publications include General Medicine (rank 1—70 609 works), Psychiatry and Mental health (rank 4—10 404 works), Education (rank 5—9 590 works), Computer Science Applications (rank 18—6 013 works), General Engineering (rank 20—5 942 works), Strategy and Management (rank 42—3 208 works), Law (rank 57—2 557 works), History (rank 62—2 329 works), Cultural Studies (rank 76—1 893 works), Pollution (rank 97—1 549 works), and Anthropology (rank 130—1 032 works). Looking at listed funders, we saw that the top three in terms of associated publications were the National Natural Science Foundation of China (3 506 works), followed by the (US) National Institutes of Health (2 316), the (US) National Science Foundation (1 022), the

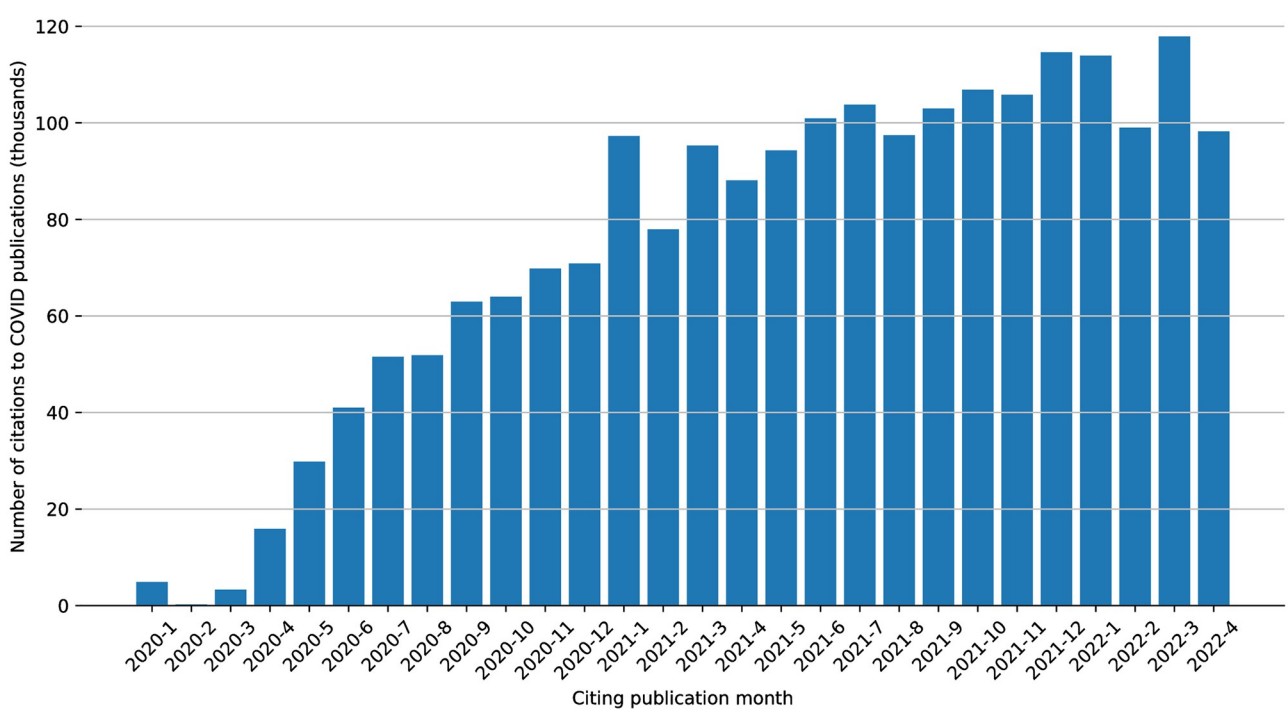

**Fig 5. Citations from COVID research to COVID research over time.**

Wellcome Trust (914), and the (UK) National Institute for Health Research (661). We also examined the affiliations of COVID study authors, propagating them to the highest parent organization (e.g. a university hospital to its university). Through this measure the top five entities were the Government of the United States of America (1 465 works), the University of California System (925), University of Toronto (910), University of London (824), and University of Oxford (660).

We also looked how long it took for COVID articles to start citing each other, building, as it where, "on shoulders of giants". As can be seen in Fig 5 a citation network ramped up relatively quickly, surpassing ten thousand citations to COVID research by April 2020, and reaching 118 thousand citations on March 2022. The large number of works published in January 2020 appears to be due to journals publishing later in the year volumes with that date. This was, for example, the case with an overview article on COVID-19 and cerebrovascular diseases [68] and an eight month retrospective [69]. (The same phenomenon may also explain the January 2021 rise.) Notably, this backdating practice can distort the establishment of priority over scientific advances based on a journal's publication date.

Finally, we examined the heightened role of preprints in COVID-19 publications [70]. In the COVID data set from the 491 945 published works, 48 211 (9.8%) were identified as preprints. Contrast this with all the works published in the same period, where from the 16 369 968 published works, 572 200 (3.5%) were identified as preprints.

## Impact and productivity metrics

We replicated diverse impact and productivity metrics typically calculated in a proprietary fashion by commercial bibliometric data providers.

The Journal Impact Factor (JIF) and Hirsh's h-index are often used as (imperfect) citation-based metrics of a venue's quality and impact [71, 72]. We calculated the 2021 JIF, and compared the 600 journals with the highest JIF available through Clarivate with 581 we matched through their ISSNs, obtaining a Spearman rank-order correlation coefficient of 0.72 with $P < 0.001$. We also calculated the journal h5-index, and compared the values of the 100 "Top publications" venues listed in Google Scholar [73] against those of Alexandria3k by hand-matching the venue titles. The comparison of the 91 matched elements gave a Spearman rank-order correlation coefficient of 0.75 with $P < 0.001$. Examining the same metric in a field where considerable work is published in conferences, we compared the h5-index of the 13 common venues between the 20 Google Scholar reports in the "Software Systems" category [74] against a curated list of 32 software engineering venues [75], and obtained a Spearman rank-order correlation coefficient of 0.83 with $P < 0.001$.

Through the same data we also obtained the most cited articles published in the corresponding period (predictably, the top-one, with 21 426 citations, was on COVID-19) [76] and overall (the top one was a 1996 article on the generalized gradient approximation method [77], which received 39 715 citations from a surprising diversity of fields). A Clarivate Web of Science query performed on May 20 2023 found the same articles ranked 5th and 4th correspondingly in terms of citation numbers.

A widely-cited study has reported a citation advantage of open access articles [78], though followup studies drew a more nuanced picture [79]. We looked at the corresponding advantage of articles published in open access journals. A comparison of the population of citation numbers to articles published from 2011 onward in open access journals and corresponding ones to articles that were not gives a negligible effect size (Cliff's $d = 0.02$; Cohen's $d = -0.035$), and also indicates that publications in non open access journals are cited more (mean 8.6) than the ones that are published in open access ones (mean 7.1). This finding is in line with the one reported by a longitudinal study that found citations associated with journals funded by article processing charges to have reached the same level as subscription journals, but to be lower for journals funded by other means [80]. On the other hand, a later study examining the relative influence of a journal's impact factor reports open access journal publications to have significantly more citations overall compared to the rest [81], so clearly more research is needed on this front.

Hirsh's h-index was originally proposed as a metric to quantify an individual's scientific research output [82]. We demonstrated this use of Alexandria3k by obtaining the h5-index on authors identified through ORCID. A noteworthy observation was the number of authors with a high metric: the top-ranked author has an h5-index of 76, twelve authors have an h5-index larger than 60, and 100 larger than 38. These achievements appear to be even more difficult to explain than earlier observations of hyperprolific authors [83]. We explored the phenomenal productivity and impact exhibited by the authors at the top of the distribution by examining the clustering coefficient of the graph induced by incoming and outgoing citations of distance 2 for a given work. The clustering coefficients of a random sample of 50 works from authors with an h5-index larger than 50 (median coefficient 0.05) appear to be significantly different from a random sample of other works of the same size with the same number of citations for each one (median 0.03). Specifically, comparing the two populations we obtained a Mann-Whitney U statistic measure of 781 for the coefficients of the top-ranked authors' works with $P < 0.001$.

## Discussion

Alexandria3k builds on a rich and evolving ecosystem of publicly available data and sophisticated open source software libraries coupled with exponential advances in computing power,

enabling scientists to perform reproducible bibliometric, scientometric, and research synthesis studies in a transparent and repeatable manner on a personal computer. In our use of Alexandria3k we found that over time its features and interface, as well as the methods we used to conduct the proof-of-concept studies, crystallized into a form we employed for distributing the code of all the proof-of-concept studies. Other researchers can readily build upon these examples to conduct their own studies.

Alexandria3k is not a panacea for the issues we identified in the introduction. Crossref's coverage of citation links is lacking compared to commercial alternatives [84]. The linkage of publications to their authors through ORCID is thin, and ORCID author metadata are also sparse. The string-based matching used to link textual author affiliations to RORs can result in mismatches. The subjects associated with works are derived from the categorization of whole journals by Scopus, and they are therefore incomplete and potentially inaccurate. The absolute values of derived bibliometric measures do not match the proprietary ones, which employ different methods for data collection (web crawling for Google Scholar) and for processing (hand-curation for Clarivate). In general, many of the supported data elements can be used for making statistical observations, but the validity of any findings needs to be verified, e.g. through sensitivity analysis based on the manual examination of a sample. Finally, one should also take into account the epistemological shortcoming of structured approaches that use publication databases [85].

Nevertheless, Alexandria3k opens many possibilities to conduct research that goes beyond what can be easily done through the query and API-based approaches offered by existing proprietary [2] and open [6] databases. Examples include the study: of collaboration patterns between organizations and disciplines, of citation cliques and organizational inbreeding, of funder, publisher, and organizational performance, of open access availability and its effects, of pre-print servers as alternatives to peer-reviewed publications, of interdisciplinary connections, and of structural publication differences between organizations or scientific fields.

Apart from making Alexandria3k available as open-source software we also structured it and intend to run it as an open-source project, accepting and integrating contributions of additional open-data sources (e.g. MEDLINE/PubMed, patent metadata, OpenAlex [6]), algorithms (e.g. publication topic classification; author name and affiliations disambiguation; citation matching), and example studies. Some of the data sources, such as PubMed and OpenAlex, overlap with Crossref, but can be useful for the additional metadata they provide, such as the MEDLINE medical subject headings (MeSH) and the OpenAlex concepts. A practical approach for integrating such data may be to create data sources that only import the metadata and link them with the *works* table using the DOI.

As part of this extension effort, Alexandria3k is participating in the 2023 Google Summer of Code initiative—a global program promoting open-source software contributions by student developers—through the Greek Open Technology Alliance—an non-profit organization with 37 Greek universities and research centers as members. We hope that running Alexandria3k as an open source project will allow it to grow organically serving ever more needs of research synthesis and analysis studies.

## Methods

The paragraphs below describe key elements behind the implementation of Alexandria3k and the reported proof-of-concept studies. The complete implementation and all the proof-of-concept studies are made available as open source software [86], and can be further examined or run on the openly available data that Alexandria3k uses [43–46] and the associated replication package [87] to obtain additional details.

## Implementation

Alexandria3k is designed around modules that handle the Crossref, ORCID, ROR, and flat file (journal names, funders, open access journals) data sources. Each module defines the data source's schema in terms of tables and their columns. The schema's SQL DDL implementation is used to define the corresponding tables in a populated database. The schema is also used for analyzing and satisfying vertical slicing requests when populating databases with Crossref and ORCID data, and for running Crossref queries without populating a database. Where possible, data are on-the-fly decompressed, extracted from an archive (with a tar or zip structure), and parsed in chunks, thus avoiding the storage cost of the entire decompressed data set (1 TB for Crossref and 467 GB for ORCID).

In its simplest form Alexandria3k can evaluate an SQL query directly on the Crossref dataset, often to perform exploratory data analysis. Results can be saved as a CSV (comma-separated values) file or iterated over through Python code for further processing. Both of these modes have limitations in terms of performance, aggregation of query results, and combination of data from multiple sources.

In most cases Alexandria3k is used to populate an SQLite database [88]. Multiple alternatives were considered for the data back-end. The native Crossref tree format would suggest a JSON-style NoSQL database. Redis [89] would be an efficient back-end, but would limit the amount of data to that that could be processed in memory. MongoGB [90] would address the capacity limitation, at the cost of a more complex installation process due to licensing issues. A client-server relational database system could also be used to offer improved integrity constraints and more advanced query optimization facilities. However, these options require the installation, configuration, connection, and maintenance of a separate database process, which is not a trivial task, especially for researchers outside the computing field. Consequently, the adopted approach uses the SQLite database, which is embedded into Alexandria3k, directly available in Python, easily installable as a command-line tool in all popular computing platforms, and accessible through APIs for diverse programming languages and environments. Transferring a database between computers only involves copying the corresponding file. Consequently, there is no need to setup, configure, and maintain a complex client-server relational or NoSQL database management system. Despite its -Lite suffix, SQLite supports large parts of standard SQL [91] (including window functions and recursive queries), and employs sophisticated query optimization methods; these feature made it ideal for use in Alexandria3k. SQLite's main downsides—lack of multi-user and client-server support—are not relevant to common Alexandria3k use cases.

Direct queries on the Crossref data set (without populating a database) are implemented by defining SQLite virtual tables that correspond to the offered schema through the Python *apsw* module. Crossref data are distributed in the form of about 26 thousand compressed containers. Running a query on them is trivial when the query accesses a single table: the Alexandria3k virtual table implementation moves from one container to the next as the table is scanned sequentially.

Direct Crossref queries involving multiple tables are handled differently. First, a dummy query execution is traced to determine the tables and fields it requires. Then, in-memory temporary tables are populated with the required data from each container, and the query is executed repeatedly on the instantiated tables. This approach works for all cases where relational joins happen within a Crossref container (e.g. works with their authors or references). More complex cases, such as relational joins between works or aggregations, require the population of an external database.

The database population design uses two techniques to improve its performance. First, records are bulk-inserted in batches by attaching directly the virtual tables to the database to be

populated, and by having the database engine perform the insert operations with a single internal command for each batch. This avoids a round-trip cost of obtaining the data in Alexandria3k and then storing them back in the database. Second, database indices over the containers in which the data are split are implemented and used so as to access each container file in turn for populating all required tables. The correspondingly improved locality of reference [92] is then utilized by caching the decompressed and parsed file contents. Unfortunately, multithreading cannot be easily used to further improve performance because both Python and SQLite make it difficult to obtain the required concurrency.

The performant matching of author affiliations with RORs is based on multiple applications of the Aho-Corasick string-matching algorithm [93]. First, an Aho-Corasick automaton is created with all unique organization names, aliases, and acronyms. Second, the automaton is used to find entries in it that also match other entries (e.g. "ai", the acronym of the "AI Corporation", also matches the organization name "Ministry of Foreign Affairs"), and mark them for removal to avoid ambiguous matches. Finally, a new automaton, constructed from the cleaned-up entries, is used to find the longest match associated with each author affiliation string. This is stored in the database as the affiliation's organization identifier. When the Alexandria3k user specifies that affiliations should match the ultimate parent organization, a recursive SQL query adds a "generation" number to each matched organization, an SQL window (analytic) function [94] orders results by generation, and a final selection query obtains the ROR identifier associated with the most senior generation.

## Proof-of-concept studies framework

We structured most proof-of-concept studies we presented as a series of queries that build on each other. This aids comprehensibility, testability, analysability, and recoverability. We specified the corresponding workflow using Makefiles [49] based on the *simple-rolap* system, which manages relational online analytical processing tasks [50, 51]. The *simple-rolap* system establishes the dependencies between queries and executes them in the required order. Most studies start with a population phase, which fills a database with the required horizontal and vertical data slices. In many cases, we used the *rdbunit* SQL unit-testing framework to test SQL queries [51]. We employed a shared Makefile with rules and configurations that satisfy dependencies required by more than one study.

In total the proof-of-concept studies distributed as examples with Alexandria3k comprise 107 SQL query files amounting to 1 876 (commented) lines of code. The query operations are organized by 18 Makefiles (511 lines), which populate the databases and execute the queries in the required order. The queries create 42 intermediate tables and facilitate their efficient operation through the creation of 48 table indices. The charts, tables, and numbers in this report are based on data produced by SQL queries that obtain their results from intermediate tables, from populated databases, or directly from the Crossref data set.

We created a vertical slice of the complete Crossref database, which we used for a number of purposes. The slice contains mainly the primary and foreign keys (including DOIs, and ORCIDs) of all entities, plus the publication year, author affiliation names, and work subjects, which are not normalized. We also ran Alexandria3k to populate the database with the Scopus All Science Journal Classification Codes (ASJCs) [95] and RORs, and linked work subjects to ASJCs and author affiliations to RORs.

We used the graph database to extract most of the database contents metrics provided in the main text and in Tables 1 and 2. This was done through simple `SELECT Count(*) FROM table` or `SELECT Count(DISTINCT id) FROM table` SQL queries.

## Scientific publishing evolution

In common with other studies [55, 96] we limited our examination to works published after World War II, in order to avoid misleading comparisons with the markedly different scientific and publication environment that preceded it. We calculated most numbers used for plotting Fig 2 from the populated Crossref graph database. We obtained the number of published works and journals through SQL `Count` aggregations of the underlying data grouped by year. We obtained the ratios of authors per work and references per work by counting the corresponding elements of the associated detail tables and then obtaining SQL `Avg` aggregations grouped by year. We joined papers with their citations using the document's DOI as a key and used this to calculate the two-year impact factor, the received citations per work, the twenty-year impact factor, and the proportion of all published works cited at least once each year. a twenty-year rolling sum over the number of works published each year. (To facilitate the required case-insensitive DOI matching, Alexandria3k converts internally all DOIs into lowercase.)

We calculated the number of pages per work by populating a database with the required work and author details, and by extracting the starting and ending page number from Crossref works that contain a dash in their `pages` field. This process excludes single-page works reported only with a single page number (rather than as a range with the same starting and ending page). We excluded from the data records with a starting page lower or equal to the ending page or those having more than 1000 pages, because the latter (164387 records) were often derived from data-entry mistakes, such as repeated page digits (e.g. `234-2366`), as well as unusable data formats, such as `1744-8069-5-32`.

We calculated a measure that can be used to track author productivity (works per author) despite clashes in author names, by taking advantage of the fact that we display productivity in relative terms (adjusting it to be 1 in 1950). In absolute terms authors with same names increase the productivity's absolute value. (An author named Smith, Kim, or Zhu would appear extremely prolific.) However, assuming that the ratio of clashing names in the population does not change, the effect of duplicate names on the relative productivity measurements is cancelled out.

Note that an actual study of author productivity would need to test and control for the assumption we made, because the population's composition might change over the years to include authors from ethnic backgrounds with more or fewer common name clashes, (for example, about 80% of China's population shares the 100 most common Chinese surnames [97]) making the phenomenon more frequent or less frequent over time. Using Alexandria3k we obtained frequently-occurring names at the two ends of the examined period and found that in 1950 the three most frequent names were W. Beinhoff (161 works—0.10% of the total names), E. Rosenberg (149—0.09%), and F. De Quervain (115—0.07%); whereas in 2021 they were Y. Wang (63363—0.23%), Y. Zhang (57414—0.21%), and Y. Li (51792—0.19%). The different percentages at the two periods' ends indicate that further controls for this change would be required, e.g. by measuring name clashes through ORCIDs while also considering variations in their adoption [98].

We calculated the $CD_5$ index [99] of Crossref publications by populating a database with their publication date and DOI, as well as the DOIs of the corresponding references. Given the available data, we were able to calculate the $CD_5$ index for six additional years (until 2016) compared to previous results [55], using the remaining five complete years we had at out disposal (2017–2021) to obtain the required citations' window.

The $CD_5$ calculation proved to be computationally challenging. The processing for the already-published 1945–2010 range, taking into account even publications lacking reference data, required more than five days of computing and about 40 GB of RAM. Surprisingly, extending the range to 2016 increased the required time to 45 days. To address this we

enhanced the original CD calculation algorithm implementation converting it to C++ [100], so at to use more efficient data structures and algorithms as well as multithreading. This allowed us to perform the $CD_5$ index calculation in 33 hours of elapsed time using 40 hours of CPU time and 49 GB of main memory.

To help other researchers build on this data without incurring the associated high computational cost, we have made the resulting data set containing the DOI and the $CD_5$ index for 50 937 400 publications in the range 1945–2016 openly available [101]. This improves upon previously available data [102], which extends to 2010 and only provides the time-stamp of each publication, without other uniquely distinguishing publication identifiers.

We calculated the evolution in the number of publications across scientific fields (Fig 4) by propagating the specific fields associated with each work in the Crossref graph database to the more general containing field. For that we used as general fields the Scopus ASJCs that ended in "00", and as their sub-fields those that started with the same numeric prefix. For example, we allocated publications under the subject of "Catalysis" (1503) to "General Chemical Engineering" (1500). We then calculated total publications in 1945 and 2021, changes in the percentages of a field's publications in terms of the total at the two time points, and included in the Figure the ten fields with the largest change whose publications amounted to more than 2% of the 2021 total.

We calculated the numbers associated with research synthesis studies by processing the output of Alexandria3k run on the Crossref data with an SQL query that matches specific words in publication titles. The query's terms are structured to give precedence to the characterization of titles indicating a systematic review as such, classifying the rest as (unspecified) secondary studies. In Fig 3 we combined the plotting of bibliometric (BM) and scientometric (SM) studies, and did not plot figures for mapping reviews (MR), umbrella reviews (UR), and tertiary studies (S3)—6061 publications in total. We also did not plot the 400 identified studies published before 1971 as well as studies published after 2021, as only part of the year 2022 data were available. Note that works containing "bibliometric" or "scientometric" in their title may either employ these methods [103] or refer to them [104]. We used another query with the same terms to list the 30 studies published before 1950 and obtain the earliest one [56] among them.

## COVID-19 metrics

To study COVID-19 publications we populated a database with a full horizontal slice of the Crossref data by specifying as the row selection criterion works containing the string "COVID" in their title or abstract ("covid" is not part of any English word). We also linked works to their subjects and author affiliations to the corresponding RORs. We obtained organizations publishing COVID-19 research by assigning author affiliations to works, and by counting both ROR-matched affiliations and unmatched affiliations as simple text.

We calculated the approximate number of researchers who worked on all COVID-19 studies by starting with the number of unique (author given-name, author surname) pairs in the set of all COVID study authors $N_{an}$. The number of true authors could be higher if many authors share the same name or lower if the same author appears differently (e.g. through the use of initials) in some publications. We address this by obtaining from the set of authors with an ORCID the number of distinct ORCIDs $N_o$, which is the true number of authors in that set, and the number of distinct names $N_{on}$, which approximates any bias also found in $N_{an}$. We then consider the true number of authors as

$$N_{an} \frac{N_o}{N_{on}}$$

Note that the employed method's accuracy is still vulnerable to ghost and empty ORCIDs that seem to be created by paper mills [105].

## Journal impact factor

We calculated the 2021 Journal Impact Factor [71] by populating a database with the keys, ISSNs, publications years, and pages of works and their references published between 2019 and 2021. We then created a table associating works with journal ISSNs. From this table we obtained citations published in 2021 to works published in 2019–2020 (the impact factor's numerator). We further filtered works to identify "citable" items, which Clarivate defines as those that make a substantial contribution to science and therefore do not include elements such as editorials and letters. For this we used as a rough heuristic works longer than two pages. (We also included works lacking a page range.) From the count of citable items per journal we obtained the number of publications published in the 2019–2020 period (the denominator). Finally, to compare our results with the numbers published by Clarivate we associated each impact factor metric with all ISSNs known for a journal (electronic, print, alternative), excluding the "alternative" ISSNs of one journal used as primary for another journal.

## Open access publications

We calculated the yearly evolution of open access journal articles and their ratio to all journal articles by using the Directory of Open Access Journals (DOAJ) provided through Alexandria3k. We joined the Crossref works with DOAJ based on each journal's ISSN to create a database table containing works published in open access journals after each journal's open access start date. We then joined the open access papers with their Crossref metadata and grouped them by year to obtain the yearly evolution shown in Fig 2. We used the set difference between all journal papers and the previously obtained joined table of the ones published in open access journals to obtain the corresponding ratio. We did not employ the Mann-Whitney U statistic for comparing the two populations, because the Shapiro-Wilk test on them indicated that the citation numbers are not normally distributed (W = 0.115, $P < 0.001$).

## Productivity metrics

We obtained the h5-index [82] productivity metrics by populating databases with data sliced vertically to include the keys of works and references and horizontally to include items published in the period 2017–2021. For the software engineering venue metrics we selected the examined conferences based on the DOI prefix assigned to the conference publications each year.

To study the citation graph of top-ranked authors we obtained a) a random sample of 50 works written by top-ranked authors, and b) a random sample of 50 works from all other publications, paired with the ones selected from the top-ranked authors to have the same number of citations as them. For each publication in the two samples, we created a separate graph containing the work $w$, the set $S$ of the works $w$ cites and the works that cite $w$, and then again the set $S'$ of the works $w'$ that cite or are cited by $w \in S$. The graph's edges are citations from one work to another. We created each citation-induced graph with a Python program querying the populated database and employed the *NetworkX* [106] Python package to calculate the graph's average clustering.

## Performance and map of available code

Performance details of the tasks reported here are summarized in Table 3. The reported figures are associated with two parts of each analysis. First, the population of an SQLite database with a horizontal and vertical slice of Crossref (and other) data. The size of this database (together with any indexes created in it) is reported as $S_{pop}$ and the corresponding time required to populate the database as $T_{pop}$. Second, the execution of ROLAP SQL queries to obtain the required results. In many cases the queries generate secondary tables, which are stored in a separate ROLAP (analysis) database. The size of the ROLAP database is reported as $S_{ROLAP}$ and the time required to run the queries as $T_{Qtot}$, $T_{Qmean}$, and $T_{Qmax}$. The query run time includes time for creating indexes, intermediate tables, and final reports. For tasks where no population figures are listed, the queries are run directly on the Crossref containers. For tasks where no ROLAP size is listed, the queries generate the results directly from the populated database, without creating any intermediate tables. All reported times are elapsed (wall clock) times shown in hours:minutes:seconds format.

The numbers shown on the table involve executions on idle or lightly loaded hosts with processes taking up a single core. The memory use (maximum resident set size—RSS) of population and query tasks averaged 1.2 GiB (minimum 4 MiB, maximum 49 GiB, median 35 MiB, $\sigma = 5$ GiB).

In addition to the SQL query times reported in the table, Python programs were used for analyzing the author graph (0:08:38 elapsed time, 55 MiB RSS) and the open source journal impact (0:01:19 elapsed time, 3.5 GiB RSS). Furthermore, a multithreading C++ program was used to calculate the $CD_5$ (9:24:20 elapsed time on 8 cores, 66:48:37 CPU time, 49 GiB RSS).

The task names in Table 3 mirror the contents of the Alexandria3k source code distribution `examples` directory. They are associated with what is reported in this work as follows. The *author-productivity* and *yearly-numpages* tasks were used to derive the corresponding yearly

**Table 3. Performance summary.**

| Task | H | $S_{pop}$ | $T_{pop}$ | $S_{ROLAP}$ | # Q | $T_{Qtot}$ | $T_{Qmean}$ | $T_{Qmax}$ |
|---|---|---|---|---|---|---|---|---|
| author-productivity | R | 18 GiB | 4:23:09 | 12 KiB | 4 | 0:24:49 | 0:06:12 | 0:22:11 |
| cdindex | S | 90 GiB | 10:59:08 | 3 GiB | 3 | 2:24:09 | 0:48:03 | 2:23:44 |
| covid | R | 3 GiB | 7:53:45 | — | 8 | 3:21:12 | 0:25:09 | 3:20:23 |
| crossref-standalone | T | — | — | — | 3 | 11:50:21 | 3:56:47 | 3:58:36 |
| graph | R | 189 GiB | 6:35:53 | 283 MiB | 37 | 19:18:29 | 0:31:18 | 11:57:56 |
| impact-factor-2021 | T | 30 GiB | 7:11:05 | 2 GiB | 12 | 0:38:54 | 0:03:14 | 0:22:11 |
| journal-h5 | S | 33 GiB | 20:41:03 | 5 GiB | 4 | 0:43:04 | 0:10:46 | 0:34:13 |
| open-access | R | 95 GiB | 9:35:37 | 555 MiB | 3 | 0:01:59 | 0:00:39 | 0:00:58 |
| orcid | T | 6 GiB | 3:49:02 | — | 1 | 0:00:02 | 0:00:02 | 0:00:02 |
| person-h5 | T | 50 GiB | 8:31:05 | 5 GiB | 7 | 0:44:12 | 0:06:18 | 0:29:15 |
| research-synthesis | S | — | — | — | 1 | 7:01:52 | 7:01:52 | 7:01:52 |
| soft-eng-h5 | T | 137 MiB | 9:45:14 | 7 MiB | 5 | 0:00:01 | 0:00:00 | 0:00:01 |
| yearly-numpages | T | 4 GiB | 4:08:07 | 2 GiB | 6 | 0:08:52 | 0:01:28 | 0:03:44 |

H(ost) R: Intel i7–10700 CPU @ 2.90 GHz, 16 MiB cache, 32 GiB DDR4 RAM, SSD storage;

H(ost) S: Intel E5–1410 CPU @ 2.80 GHz, 10 MiB cache, 72 GiB DDR3 RAM, magnetic disk;

H(ost) T: Intel i7–7700 CPU @ 3.60 GHz, 8 MiB cache, 16 GiB DDR4 RAM, magnetic disk;

$S_{pop}$: Populated database size; $T_{pop}$ Time to populate database;

$S_{ROLAP}$: Relational analytical processing database size; # Q: number of queries run;

$T_{Qtot}$: total query time; $T_{Qmean}$: average (arithmetic) query time; $T_{Qmax}$: maximum query time

evolution lines shown in Fig 2. The *crossref-standalone* task derived the evolution in the number of journals (Fig 2), the yearly availability of abstracts (Fig 1), and the types of works available in Crossref. The *open-access* task derived the evolution in the number of papers published in open access journals (Fig 2), the reported ratio, and the statistical analysis regarding citations to publications in open access journals. The *graph* task was used to derive the remaining yearly evolution and yearly availability metrics, the Crossref record metrics (Table 1), and the evolution of publications in scientific fields (Fig 4). The *orcid* task derived the ORCID metrics shown in Table 2. The *research-synthesis* task derived the evolution in systematic literature reviews (Fig 3). Finally, *cdindex* derived the $CD_5$ index, *covid* the COVID-19 figures, *impact-factor-2021* the 2021 journal impact factor, *journal-h5* the journal h5-index, *person-h5* the person h5-index, and *soft-eng-h5* the h5-index of software engineering venues.

## Statistical analysis

For reporting the correlation between the metrics obtained by Alexandria3k and existing ones and for comparing the graph clustering coefficients between two populations we used the functions *spearmanr* and *mannwhitneyu* from the Python package *scipy.stats*. All calculations were performed with "two-sided" as the alternative hypothesis (the default). No other options were provided to the function calls. We calculated CAGR with *LinearRegression* function of the *sklearn.linear_model* package. For the analysis and charting we used Python 3.9.10 with the packages *cliffs-delta* 1.0.0, *matplotlib* 3.3.4, *numpy* 1.20.1, *pandas* 1.2.4, *scikit-learn* 0.24.1, *pySankey* 0.0.1, and *scipy* 1.6.2.

## Supporting information

**S1 Fig. Relational schema of Crossref tables in a fully-populated database.**
(PDF)

**S2 Fig. Relational schema of ORCID tables in a fully-populated database.**
(PDF)

**S3 Fig. Relational schema of ROR tables in a fully-populated database.**
(PDF)

**S4 Fig. Relational schema of other tables and Alexandria3k-generated links in a fully-populated database.**
(PDF)

## Acknowledgments

The author thanks Panos Louridas, Arie van Deursen, Theodoros Evgeniou, Alberto Bacchelli, Dirk Beyer, and Dimitris Karlis for valuable advice and feedback. Support from the Google Summer of Code program and the Open Technologies Alliance is gratefully acknowledged.

## Author Contributions

**Conceptualization:** Diomidis Spinellis.

**Data curation:** Diomidis Spinellis.

**Funding acquisition:** Diomidis Spinellis.

**Investigation:** Diomidis Spinellis.

**Methodology:** Diomidis Spinellis.

**Project administration:** Diomidis Spinellis.

**Resources:** Diomidis Spinellis.

**Software:** Diomidis Spinellis.

**Validation:** Diomidis Spinellis.

**Visualization:** Diomidis Spinellis.

**Writing – original draft:** Diomidis Spinellis.

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
