## [Decision Letter · Decision Letter 0]

14 Jul 2023

PONE-D-23-15521Open reproducible publication researchPLOS ONE

Dear Dr. Spinellis,

Thank you for submitting your manuscript to PLOS ONE. After careful consideration, we feel that it has merit but does not fully meet PLOS ONE’s publication criteria as it currently stands. Therefore, we invite you to submit a revised version of the manuscript that addresses the points raised during the review process.

Both reviewers agree on the merit of your paper, by suggesting improvements in terms of readability and documentation. Consider carefully their suggestion for a revised version. Please submit your revised manuscript by Aug 28 2023 11:59PM. If you will need more time than this to complete your revisions, please reply to this message or contact the journal office at plosone@plos.org. Please include the following items when submitting your revised manuscript:A rebuttal letter that responds to each point raised by the academic editor and reviewer(s). You should upload this letter as a separate file labeled 'Response to Reviewers'.A marked-up copy of your manuscript that highlights changes made to the original version. You should upload this as a separate file labeled 'Revised Manuscript with Track Changes'.An unmarked version of your revised paper without tracked changes. You should upload this as a separate file labeled 'Manuscript'.

We look forward to receiving your revised manuscript.

Kind regards,

Alberto Baccini, Ph.D.

Academic Editor

PLOS ONE

“Google Summer of Code through the Open Technologies Alliance supported the work of a student developer on Alexandria3k. The author has declared that no competing interests exist.”

Reviewers' comments:

Reviewer's Responses to Questions

**Comments to the Author**

1. Is the manuscript technically sound, and do the data support the conclusions?

Reviewer #1: Yes

Reviewer #2: Yes

2. Has the statistical analysis been performed appropriately and rigorously? 

Reviewer #1: Yes

Reviewer #2: Yes

3. Have the authors made all data underlying the findings in their manuscript fully available?

Reviewer #1: Yes

Reviewer #2: Yes

4. Is the manuscript presented in an intelligible fashion and written in standard English?

Reviewer #1: Yes

Reviewer #2: No

5. Review Comments to the Author

Reviewer #1: This manuscript tackles the issue of reproducible scientometric studies. It introduces the Alexandria3k Python software package that is showcased on a case study of COVID-19 research.

The paper is very well organised and meticulously written. It manages to motivate the need for reproducible research in scientometrics and introduces the features of Alexandria3k using laymen terms while providing technical details that will inform specialists of database design. I commend the author for stressing the limitations of his case study relying on open yet incomplete data, in the Discussion section.

I recommend ‘Minor Accept’ and welcome the author to consider the following comments to revise the manuscript:

* C1. The title seems not specific enough

I believe the title should better reflect the contribution by referring to scientometrics and Alexandria3k. A tentative title:

Open reproducible scientometric research with Alexandria3k

* C2. Mention of downstream software for bibliometric data analysis and visualisation

I kept thinking of bibliometrix (https://www.bibliometrix.org) while reading the submission. I believe readers would benefit from a paragraph explaining how the data harvested by Alexandria3k could feed bibliometrix for further data foraging.

Aria, M. & Cuccurullo, C. (2017). bibliometrix: An R-tool for comprehensive science mapping analysis, Journal of Informetrics, 11(4), pp 959-975, Elsevier, DOI: 10.1016/j.joi.2017.08.007

* C3. OpenAlex as a potential source

The author cites OpenAlex on page 4, mentioning that pySciSci sources from OpenAlex. I understand that OpenAlex aggregates and cleanses data from various bibliographic data providers, such as Crossref. Would if make sense to feed Alexadria3k with the OpenAlex snapshot (https://docs.openalex.org/download-all-data/openalex-snapshot)? Could the author consider discussing this so that authors learn about the pros and cons of this data ingestion strategy?

* C4. Clarification regarding COVID-19 publications

On page 12, the authors mentions COVID-19 publications. I wondered about the share of preprints vs journal article in the harvested dataset, since preprints were so instrumental to the COVID-19 research.

* C5. Potential use of multithreading

p18: “the evaluation is evaluated sequentially on each Crossref container” -> could the author discuss the pros and cons of multithreading to perform parallel data loading?

* Misc

- p13: “with a p-value 3 × 10^{−93}” (and elsewhere) -> p < 0.001, see the submission guidelines (https://journals.plos.org/plosone/s/submission-guidelines.#loc-statistical-reporting) and comments on social media (https://twitter.com/hippopedoid/status/1673806261895266306/retweets/with_comments).

- p17: “MondGB” -> MongoDB

- p21: “DOI as a key” -> since DOI resolution is case insensitive, it might be worth specifying that all DOIs where lowercased prior to any matching process -- if that's the case indeed.

- p21: “by extracting the starting and ending page number [...] We excluded from the data records with...”. Did the author exclude records featuring pageStart > pageEnd?

- p25: “the true number of authors” in ORCID. The author might appreciate reading:

Teixeira da Silva, J. A. (2021). Abuse of ORCID’s weaknesses by authors who use paper mills. In Scientometrics (Vol. 126, Issue 7, pp. 6119–6125). https://doi.org/10.1007/s11192-021-03996-x

- As a complement to [88] on Chinese surnames, the author might appreciate reading:

Youtie, J., Carley, S., Porter, A. L., & Shapira, P. (2017). Tracking researchers and their outputs: new insights from ORCIDs. In Scientometrics (Vol. 113, Issue 1, pp. 437–453). https://doi.org/10.1007/s11192-017-2473-0

- The author could consider replacing [31] by:

Hendricks, G., Tkaczyk, D., Lin, J., & Feeney, P. (2020). Crossref: The sustainable source of community-owned scholarly metadata. In Quantitative Science Studies (Vol. 1, Issue 1, pp. 414–427). https://doi.org/10.1162/qss_a_00022

Reviewer #2: The paper presents Alexandria3k, a Python package that can populate relational databases with data from several open publication metadata datasets, including CrossRef, ORCID, and ROR. Alexandria3k enables scientists and science analysts to perform reproducible quantitative studies of science without recurring to the usual commercial products, such as Web of Science or Scopus.

Even if I am sympathetic with this project and I think Alexandria3k might be a useful tool to add to the toolbox of science analysts, I must confess that I am a bit pessimistic about its capacity to realistically compete with commercial products that can rely on massive financial resources and do not have many of the limitations of open-access datasets such as CrossRef. That said, Alexandria3k is in my opinion technically sound and can be useful, especially to members of the scientific community that cannot access commercial databases.

My main concern is the overall structure of the paper. In its present form, it is very long and presents too much material to the readers. I would suggest to the Author(s) to shorten the paper significantly and reorganize its content in order to enhance the clarity of the presentation. For instance, the paper begins in medias res, so to say, by presenting a result obtained with Alexandria3k, even before the tool is properly introduced and the problems it aims to resolve presented. In the current version of the Introduction, the readers find themselves in the middle of numerous figures and statistics about the raise of science studies without a context, leaving them confused. I would rather suggest the Author(s) to start the paper with a classic overview of the problem Alexandria3k is meant to resolve and with a description of the state of the art around the problem (i.e., expand the section that starts around row 23).

Moreover, I would suggest the Author(s) to reduce the number of proof-of-concept studies of Alexandria3k and to significantly shorten the Methods section: I would select three or four studies and present them in separate paragraphs, with research question, methods, and results of each of them clearly explained, avoiding useless technical minutiae. Further studies can be left to Supplementary Materials, added to the supporting information of the package or even presented in a separate paper.

After these changes in the structure of the paper, in my view it can proceed to publication.

6. PLOS authors have the option to publish the peer review history of their article (what does this mean?). If published, this will include your full peer review and any attached files.

Reviewer #1: No

Reviewer #2: No

---

## [Author Response · Author response to Decision Letter 0]

16 Sep 2023

Please see detailed response in the submitted "Response to Reviewers" document.

---

## [Decision Letter · Decision Letter 1]

13 Nov 2023

Open reproducible scientometric research with Alexandria3k

PONE-D-23-15521R1

Dear Dr. Spinellis,

We’re pleased to inform you that your manuscript has been judged scientifically suitable for publication and will be formally accepted for publication once it meets all outstanding technical requirements.

Kind regards,

Alberto Baccini, Ph.D.

Academic Editor

PLOS ONE

Additional Editor Comments (optional):

Reviewers' comments:

Reviewer's Responses to Questions

**Comments to the Author**

1. If the authors have adequately addressed your comments raised in a previous round of review and you feel that this manuscript is now acceptable for publication, you may indicate that here to bypass the “Comments to the Author” section, enter your conflict of interest statement in the “Confidential to Editor” section, and submit your "Accept" recommendation.

Reviewer #1: All comments have been addressed

Reviewer #2: All comments have been addressed

2. Is the manuscript technically sound, and do the data support the conclusions?

Reviewer #1: (No Response)

Reviewer #2: Yes

3. Has the statistical analysis been performed appropriately and rigorously? 

Reviewer #1: (No Response)

Reviewer #2: Yes

4. Have the authors made all data underlying the findings in their manuscript fully available?

Reviewer #1: (No Response)

Reviewer #2: Yes

5. Is the manuscript presented in an intelligible fashion and written in standard English?

Reviewer #1: (No Response)

Reviewer #2: Yes

6. Review Comments to the Author

Reviewer #1: (No Response)

Reviewer #2: All my comments were successfully addressed by the Author. In my view, the manuscript can proceed to publication.

7. PLOS authors have the option to publish the peer review history of their article (what does this mean?). If published, this will include your full peer review and any attached files.

Reviewer #1: **Yes: **Guillaume Cabanac

Reviewer #2: No

---

## [Editor Report · Acceptance letter]

17 Nov 2023

PONE-D-23-15521R1 

Open reproducible scientometric research with Alexandria3k 

Dear Dr. Spinellis:

I'm pleased to inform you that your manuscript has been deemed suitable for publication in PLOS ONE. Congratulations! Your manuscript is now with our production department. 

Kind regards, 

on behalf of

Prof. Alberto Baccini 

Academic Editor

PLOS ONE